**Article** https://doi.org/10.1038/s41467-023-40591-5

# Ionic liquid gating induced self-intercalation of transition metal chalcogenides

Fei Wang[1,4], Yang Zhang [1,4], Zhijie Wang [2,4], Haoxiong Zhang[1], Xi Wu[2], Changhua Bao[1], Jia Li [2] ✉, Pu Yu [1,3] ✉ & Shuyun Zhou [1,3] ✉

Ionic liquids provide versatile pathways for controlling the structures and properties of quantum materials. Previous studies have reported electrostatic gating of nanometer-thick flakes leading to emergent superconductivity, insertion or extraction of protons and oxygen ions in perovskite oxide films enabling the control of different phases and material properties, and intercalation of large-sized organic cations into layered crystals giving access to tailored superconductivity. Here, we report an ionic-liquid gating method to form three-dimensional transition metal monochalcogenides (TMMCs) by driving the metals dissolved from layered transition metal dichalcogenides (TMDCs) into the van der Waals gap. We demonstrate the successful self-intercalation of $PdTe_2$ and $NiTe_2$, turning them into high-quality PdTe and NiTe single crystals, respectively. Moreover, the monochalcogenides exhibit distinctive properties from dichalcogenides. For instance, the self-intercalation of $PdTe_2$ leads to the emergence of superconductivity in PdTe. Our work provides a synthesis pathway for TMMCs by means of ionic liquid gating driven self-intercalation.

Transition metal dichalcogenides (TMDCs), e.g., $PtTe_2$, $PdTe_2$, and $NiTe_2$ which are type-II Dirac semimetals[1–3], have been widely investigated, thanks to the facile availability of high-quality single crystals[1,4,5]. In contrast to TMDCs, their monochalcogenide counterparts $PtTe$[6], $PdTe$[7], and $NiTe$[8] are much more difficult to grow into single crystal form due to the large formation energy, and the as-grown samples often contain an intergrowth of Te and TMDCs[9]. While PdTe has been reported as a strongly coupled superconductor based on transport measurements on polycrystal samples one decade ago[10], single crystals of PdTe have been synthesized only recently by annealing $PdTe_2$-Pd, forming a thin layer of PdTe at the interface[7] with a superconducting transition temperature $T_C$ of 2.6 K. Intriguing properties such as superconductivity, magnetism, and topological superconductivity have also been predicted in transition metal monochalcogenides (TMMCs)[9,11], while the corresponding experimental

investigation is still highly demanded due to the lack of high-quality samples. Therefore, finding a convenient method to obtain high-quality bulk TMMC single crystals can pave an important step toward exploration of these exotic material properties.

Here, we develop a synthesis method for obtaining high-quality PdTe and NiTe single crystals via self-intercalation of the transition metal ions into $PdTe_2$ and $NiTe_2$. Specifically, the self-intercalation process is enabled by an ionic liquid gating induced electrochemical reaction, where metals from TMDCs are first dissolved by the ionic liquids and then incorporated into the van der Waals gap driven by the external electric field, forming TMMC single crystals. In addition to providing an effective method for obtaining high-quality PdTe and NiTe single crystals, our work also enriches the applications of well-explored ionic liquid gating method in controlling material structures and properties via gating-induced self-intercalation.

[1]State Key Laboratory of Low Dimensional Quantum Physics and Department of Physics, Tsinghua University, Beijing 100084, People's Republic of China. [2]Shenzhen Geim Graphene Center and Institute of Materials Research, Tsinghua Shenzhen International Graduate School, Tsinghua University, Shenzhen 518055, People's Republic of China. [3]Frontier Science Center for Quantum Information, Beijing 100084, People's Republic of China. [4]These authors contributed equally: Fei Wang, Yang Zhang, Zhijie Wang. ✉e-mail: li.jia@sz.tsinghua.edu.cn; yupu@mail.tsinghua.edu.cn; syzhou@mail.tsinghua.edu.cn

## Results and discussion

### Ionic liquid gating induced self-intercalation

Ionic liquids, which contain conductive ions at room temperature, have been widely employed as important media with versatile applications in tailoring material properties. The ionic liquid gating has been successfully employed as a convenient pathway to electrostatically inject substantial carriers (electrons or holes depending on the polarity of the external voltage) into nanometer-thick films or flakes (Fig. 1a) by constructing an electric double-layer transistor (EDLT)[12], where interfacial electron doping can lead to extremely high two-dimensional carrier density (up to $10^{14}$ cm$^{-2}$), resulting in novel phenomena such as superconductivity and quantum phase transitions[12-18]. Over the past decade, the ionic liquid gating method has been extended to manipulate the ionic intercalations into three-dimensional compounds with small ions (such as protons $H^+$ or oxygen ions $O^{2-}$) dissolved in or generated from the ionic liquids (Fig. 1b), leading to distinct structural transformation with corresponding manipulation of the electrical and magnetic properties[19-23]. More recently, the ionic evolution approach has been extended to large-sized organic cations from ionic liquids[24], such as $[C_2MIm]^+$, $[DEME]^+$, etc. These organic cations are intercalated into the van der Waals gap of layered materials, such as $MoTe_2$, $NbSe_2$ etc, forming organic-inorganic hybrid materials in the bulk form[24-29] (Fig. 1c), which exhibit distinctive properties from their bulk single crystals and monolayer counterparts. In this work, we report an application of ionic liquid gating in turning TMDCs into TMMCs through a gating-induced self-intercalation process. Here, during the ionic liquid gating, the ionic liquid electrochemically dissolves the TMDCs surface, and then drives the metal ions to intercalate into the van der Waals gap of TMDCs, forming three-dimensional TMMCs (Fig. 1d) with distinctive material properties.

### Self-intercalation of PdTe₂

The experimental setup used for the ionic liquid gating induced self-intercalation is schematically illustrated in Fig. 1d−f (see Supplementary Fig. 1 and "Methods" for details). Specifically, $PdTe_2$ single crystal is used as the initial TMDCs to testify this approach. The starting TMDC single crystal and platinum plate are connected to the negative and positive electrodes, respectively, and they are both immersed in the ionic liquid of $[C_2MIm]^+[TFSI]^-$. To boost the electrochemical reaction, the ionic liquid gating is performed at an elevated temperature of 150 °C. A systematic change of the gating voltage shows that a threshold voltage of −3.2 V is required to activate the intercalation process (see Supplementary Fig. 3a). Above the voltage threshold, the ionic liquid gating electrochemically etches the $PdTe_2$ by breaking the chemical bonds, releasing $Pd^{4+}$ and $Te^{2-}$ ions into the liquids. The dissolved $Pd^{4+}$ cations are driven by the electric field to the negative electrode (i.e., $PdTe_2$ single crystal), and the aggregation of $Pd^{4+}$ subsequently facilitates the self-intercalation of ions into the bulk compound. This is confirmed by cyclic voltametric measurements of the intercalation process (Supplementary Fig. 2). In order to optimize the

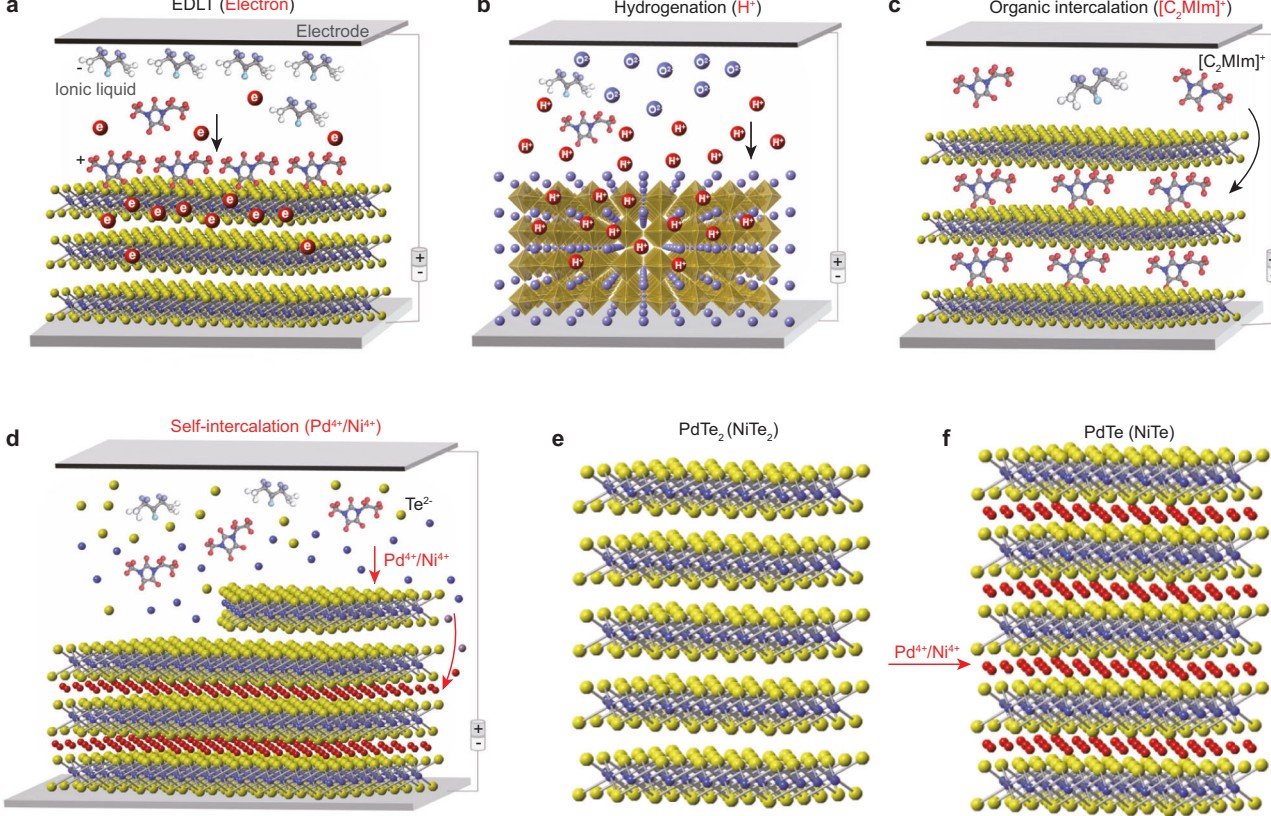

**Fig. 1 | Schematic illustrations of four different applications of ionic liquid gating in controlling the material structures and properties. a** Electrostatic doping through ionic liquid gating induced electric double-layer transistor (EDLT). The molecules represent the organic anions and cations of inoic liquid. The arrow shows the movement of electrons. **b** Ionic liquid gating induced ionic ($H^+$ or $O^{2-}$) evolution into complex oxides. The arrow shows the movement of $H^+$. **c** Organic cation intercalation driven by ionic liguid gating into layered compounds. The arrow shows the movement of organic cations ($[C_2MIm]^+$). **d** Schematic of self-intercalation process. During this process, the top layer of TMDCs is electrochemically etched by the ionic liquid gating with the anions dissolved into the ionic liquid, while the cations are intercalated into the compound. The yellow, blue and red atoms represent Te ions, Pd ions and self-intercalated Pd (Ni) ions, respectively. The arrows show the movement of Pd (Ni) ions. **e** Schematic of atomic structure for $PdTe_2$ ($NiTe_2$) before intercalation. **f** Schematic of atomic structure for PdTe (NiTe).

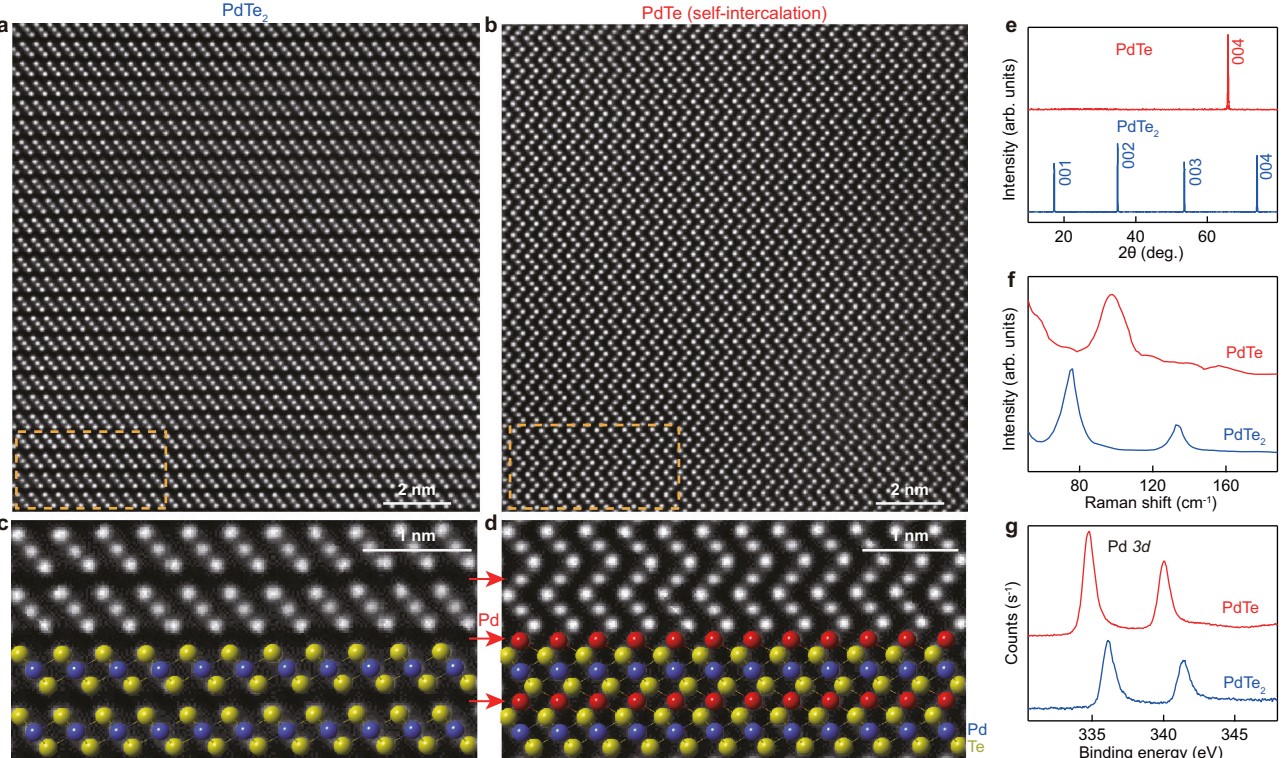

**Fig. 2 | Experimental evidences supporting self-intercalation of PdTe₂ into PdTe. a, b** Atomic-resolved high-angle annular dark field scanning transmission electron microscopy (HAADF-STEM) images of PdTe₂ before intercalation (**a**) and PdTe after intercalation (**b**). The zone axis is along the [100] of single crystal. **c, d** Zoom-in HAADF-STEM image from the orange rectangles marked in (**a, b**).

Crystal structures of PdTe₂ and PdTe are overlaid to reveal the intercalation of Pd (red symbol). **e** θ–2θ scans of PdTe₂ and PdTe in X-ray diffraction (XRD) measurements. **f** Raman spectra of PdTe₂ and PdTe. **g** X-ray photoemission spectroscopy (XPS) spectra of the Pd *3d* core level for PdTe₂ before the intercalation (blue curve) and PdTe after intercalation (red curve).

experimental conditions, ex situ X-ray diffraction (XRD), Raman spectroscopy measurements were performed during the intercalation process to monitor the extent of the intercalation at different time durations, from which the optimized experimental conditions for self-intercalation of PdTe₂ are determined to be −3.2 V at 150 °C. Supplementary Fig. 3 shows that after sufficient reaction time (more than 2 days), a complete transition of PdTe₂ into PdTe is achieved with the sample thickness of ~100 μm. Moreover, the self-intercalation has also been successfully demonstrated on exfoliated PdTe₂ flakes with thickness of 170 nm (Supplementary Fig. 4), showing that the self-intercalation applies not only to bulk crystals but also to thin flakes.

The successful structural transformation from PdTe₂ into PdTe is directly revealed by high-angle annular dark field scanning transmission electron microscopy (HAADF-STEM) images. Figure 2a, b shows a comparison of the HAADF-STEM images before and after the self-intercalation. Before intercalation, the HAADF-STEM image for the pristine PdTe₂ shows a layered structure, where sandwiches containing Te-Pd-Te layers are stacked along the vertical (c-axis) direction (Fig. 2c). After intercalation, the van der Waals gap between neighboring Te-Pd-Te layers is filled by one monolayer of Pd atoms (indicated by red symbols in Fig. 2d), which bonds with both the upper and lower layers of Te atoms and forms zigzag structures along the vertical direction (Fig. 2b, d). It is worth noting that the sharp HAADF-STEM images before and after the intercalation indicate that both crystals remain high quality.

The successful intercalation of the PdTe₂ single crystal is also confirmed by XRD measurements (Fig. 2e). After intercalation, a new set of PdTe diffraction peaks are observed without any detectable PdTe₂ peaks, suggesting that the obtained crystal is a single phase. We note that the intercalation of large-sized organic cations such as [C₂MIm]⁺, which has been reported in MoTe₂, NbSe₂, SnSe₂, etc.[24–26,28,29]

with characteristic XRD peaks at a larger interlayer spacing, is not observed in the current study, suggesting that the intercalation of large-sized organic cations from the ionic liquid is not favored in the current study. We believe that this difference could be attributed to the larger interlayer binding energy (51 meV·Å⁻²) of PdTe₂ as compared with that (~25 meV·Å⁻²) for other TMDCs[30]. Such larger interlayer binding energy does not favor the intercalation of the large-sized organic cations with dramatic lattice expansion, while the small-sized transitional metal cations can still be intercalated.

From the XRD characteristic peaks, the refined lattice parameters are extracted to be $c = 5.13 ± 0.01$ Å for PdTe₂ to $c = 5.67 ± 0.03$ Å for PdTe, consistent with previous reports[10]. This is also consistent with the lattice constants extracted by fitting the HAADF-STEM images (Supplementary Fig. 5a–d), which shows an increase of c-axis lattice constant from 5.12 to 5.67 Å, accompanied by a slight change in the in-plane lattice constant from 4.03 to 4.15 Å after the structural transition. The transition from PdTe₂ to PdTe is also supported by Raman spectroscopy measurements (Fig. 2f), where two peaks at 76.1 and 133.9 cm⁻¹ corresponding to the in-plane $E_g$ and out-of-plane $A_{1g}$ vibration modes of PdTe₂[31] are observed before the self-intercalation, while characteristic Raman peak at 98.0 cm⁻¹ of PdTe crystal[32] emerges after self-intercalation. In addition, the self-intercalation is also supported by elemental analysis in X-ray energy dispersive spectroscopy (EDS) measurements (Supplementary Fig. 5e, f) and X-ray photoemission spectroscopy (XPS) measurements (Fig. 2g and Supplementary Fig. 6), where the binding energy of Pd *3d* peak changes significantly after self-intercalation.

## Superconductivity in self-intercalated PdTe₂

The self-intercalation of Pd into PdTe₂ not only leads to a distinct structural change, but also modifies dramatically the material

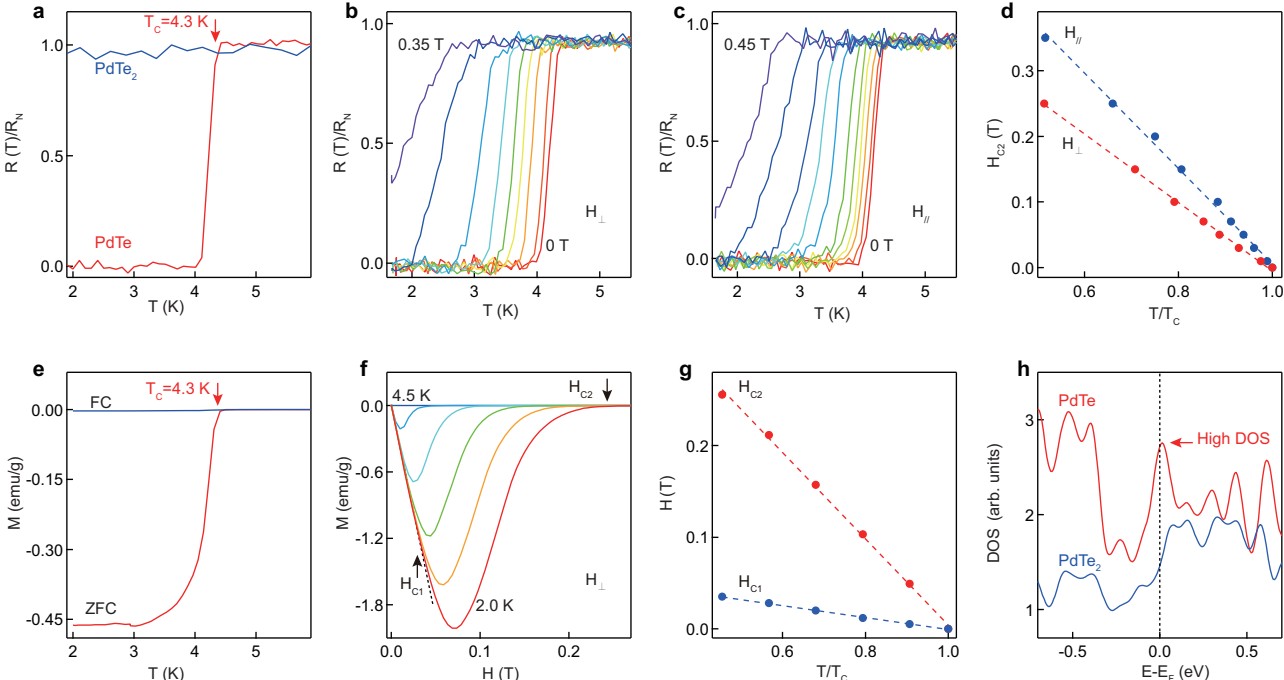

**Fig. 3 | Isotropic type-II superconductivity in PdTe. a** Temperature-dependent resistance measurements on $PdTe_2$ and PdTe. $T_C$ is defined at 90% of the normal-state resistance. The data were normalized by the resistance at 6 K ($R_N$). **b, c** Resistance of PdTe under out-of-plane and in-plane magnetic fields. The out-of-plane magnetic fields are 0, 0.01, 0.03, 0.05, 0.07, 0.1, 0.15, 0.25, 0.35 T from red to purple curve. The in-plane magnetic fields are 0, 0.01, 0.03, 0.05, 0.07, 0.1, 0.15, 0.2, 0.25, 0.35, 0.45 T from red to purple curves. The resistance were normalized. **d** Extracted upper critical magnetic fields $H_{C2}$ for $H_\perp$ and $H_{//}$ as a function of temperature. The dashed curves are fitted by Ginzburg-Landau equation. **e** DC-magnetization measured in zero-field cooled (ZFC) and field cooled (FC) conditions for PdTe with 50 Oe applied magnetic field. **f** Isothermal magnetizations (M) in the superconducting state of PdTe as a function of the applied out-of-plane magnetic field. The measurement temperatures are 2.0, 2.5, 3.0, 3.5, 4.0, 4.5 K from red to blue curves. **g** Extracted lower magnetic critical field $H_{C1}$ and upper magnetic critical field $H_{C2}$ as a function of temperature. The dashed curves are fitted by Ginzburg-Landau equation. **h** Calculated density of states (DOS) for $PdTe_2$ and PdTe. The dashed line marks the Fermi level.

properties with emergent superconductivity. While $PdTe_2$ sample does not exhibit superconductivity down to 1.8 K, superconductivity is observed in PdTe with a transition temperature of $T_C = 4.3$ K (defined at 90% of the normal-state resistance) as shown in Fig. 3a. The residual resistance ratio (RRR, defined by the ratio of the resistance at 300 K to that at 4.5 K) is extracted to be RRR = 50 for PdTe (Supplementary Fig. 7a), which is higher than the value RRR = 36 reported for polycrystalline PdTe previously[10], indicating the high quality of the PdTe sample. Figure 3b, c shows the magnetoresistance under the application of an out-of-plane (Fig. 3b) and in-plane (Fig. 3c) magnetic field, respectively, and Fig. 3d shows the extracted upper critical fields as a function of temperature. A positive magnetoresistance (MR) was observed with the transition temperature shifting gradually toward lower temperature for both the in-plane and out-of-plane upper critical fields. By fitting the in-plane and out-of-plane upper critical field $H_{C2}$ using the Ginzburg-Landau theory[33], $H_{c2}(T) = \frac{\Phi_0}{2\pi\zeta(0)^2}\left(1 - \frac{T}{T_c}\right)$, where $\Phi_0$ is the magnetic flux quantum, the in-plane and out-of-plane coherence length is extracted to be $\zeta_{//}(0) = 29.7 \pm 0.5$ nm and $\zeta_\perp(0) = 35.4 \pm 0.2$ nm, respectively. Here, the field response is much more isotropic than the reported value in PdTe films[7], where the coherence length of 25.4 nm is likely limited by the film thickness. In our case, the PdTe sample is a bulk single crystal with a thickness much larger than the coherence length, and the measured coherence length reflects the intrinsic property of the bulk material.

Magnetization measurements were also performed to reveal the superconducting properties. Figure 3e shows a clear diamagnetic response with the critical temperature of 4.3 K for both zero-field cooled (ZFC) (Supplementary Fig. 7b) and field-cooled (FC) magnetization curves, suggesting the emergence of bulk superconductivity below this temperature. From the diamagnetic response, the

superconducting shielding volume fraction is calculated to be 53%. Figure 3f shows the isothermal magnetization curves recorded from 2 to 4.5 K in the superconducting state. At 2 K, the upper critical magnetic field is 0.25 T, which is approximately twice of the value previously reported in polycrystalline PdTe samples[9,10]. Figure 3g shows the extracted lower critical field ($H_{C1}$) and upper critical field ($H_{C2}$) as a function of the normalized temperature, in which the thermodynamic critical field ($H_C$) is defined as $(H_{C1}H_{C2})^{1/2}$. In the Ginzburg-Landau theory, the upper critical field $H_{C2}$ and thermodynamic critical field $H_C$ are related by $H_{C2} = 2^{1/2}\kappa H_C$. From the extracted $H_C$ and $H_{C2}$, $\kappa$ is estimated to be 1.96, which is much larger than $(1/2)^{1/2}$, supporting that PdTe is a type-II superconductor[33]. The emergence of superconductivity in PdTe is attributed to the enhanced density of states (DOS) at the Fermi energy, which is supported by the enhancement of the DOS at the Fermi level from density functional theory (DFT) calculations (Fig. 3h). According to the Bardeen-Cooper-Schrieffer (BCS) theory, such an increase of DOS at the Fermi level indicates that more electrons are susceptible to phonon-mediated pairing interactions, leading to an increase in the superconducting transition temperature. In addition to ionic liquid of $[C_2MIm]^+[TFSI]^-$, other ionic liquids with different cations such as $[C_8MIm]^+[TFSI]^-$ (Supplementary Fig. 8a–c) and $[C_{10}MIm]^+[TFSI]^-$ (Supplementary Fig. 8d) can also be used as solvents for the self-intercalation process, and the obtained samples all exhibit superconductivity with a similar transition temperature and characteristic features.

## Self-intercalation of $NiTe_2$
The self-intercalation method can also apply to other TMDCs materials, e.g., $NiTe_2$, forming NiTe single crystals. Figure 4a–d shows HAADF-STEM images before and after self-intercalation, where a structural

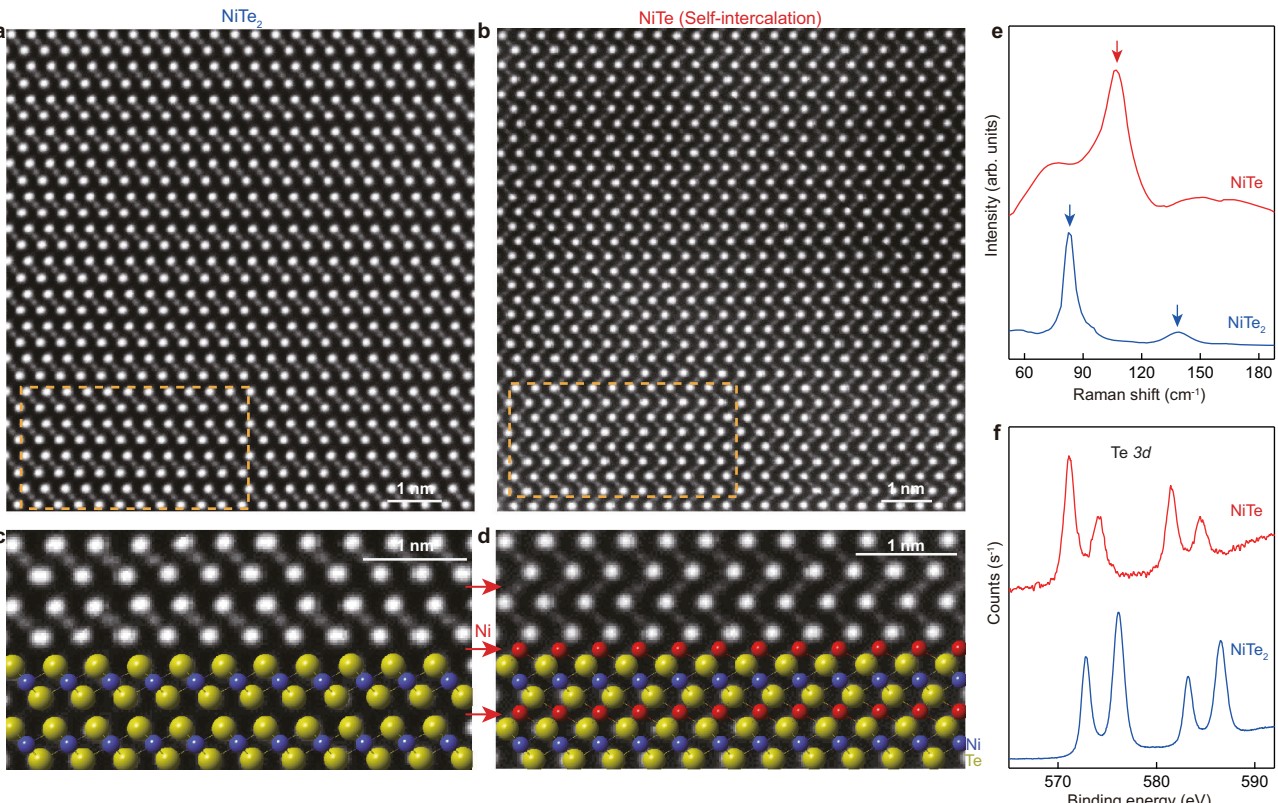

**Fig. 4 | Self-intercalation in NiTe$_2$ single crystal. a, b** Atomic-resolution HAADF-STEM image of pristine NiTe$_2$ (**a**) and self-intercalated NiTe (**b**). The zone axis is along the [100] of single crystal. **c** The magnified HAADF-STEM image from the orange rectangle in (**a**) that agrees with the overlaid atomic model structure. **d** A magnified view of HAADF-STEM image from orange rectangle in (**b**). The inset shows the atomic model structure of NiTe for comparison and the red atom in NiTe represent the intercalated Ni. **e** Raman spectra of NiTe$_2$ and NiTe. The red and blue arrows show the peak position of NiTe and NiTe$_2$, respectively. **f** XPS spectra for the Te $3d$ core level of the NiTe$_2$ and NiTe.

transition from NiTe$_2$ to NiTe is nicely observed, similar to the transition from PdTe$_2$ to PdTe discussed above. The structural change is also supported by XRD measurements and the optimized voltage to achieve the self-intercalation in NiTe$_2$ is −3.4 V at 170 (Supplementary Fig. 9). Analysis of the HAADF-STEM data shows that from NiTe$_2$ to NiTe, the a-axis lattice constant increases from 3.87 to 3.93 Å, while the c-axis increases from 5.28 to 5.37 Å (Supplementary Fig. 10a, b). To understand the intercalation process, we stopped the intercalation process of one sample before it was fully intercalated. The cross-section STEM image (Supplementary Fig. 10) of this sample shows that there is ~100-nm-thick NiTe on the surface. The EDS analysis shows a significant increase of Ni element while the Te signal remains the same, suggesting that the phase transition is caused by Ni intercalation and not by Te deficiency. Moreover, the STEM image also shows that the intercalation starts from the surface to the interior of the NiTe$_2$ single crystal, and the obtained NiTe/NiTe$_2$ hetero-junction forms an exotic material platform for future studies. In addition, we find that singe phase NiTe can be obtained by repeated exfoliation of the NiTe/NiTe$_2$ hetero-junction. The transformation of NiTe$_2$ to NiTe is also supported by Raman measurements in Fig. 4e, where a change from two characteristic Raman peaks of NiTe$_2$ to one peak for NiTe is observed. In addition, the self-intercalation also leads to a change in the valence state, as confirmed by the shift of the Te binding energy in Fig. 4f. The availability of NiTe single crystals makes it possible to investigate the predicted topological and superconducting properties at low temperature[11] in the future study.

## Mechanism of self-intercalation

In order to understand the self-intercalation mechanism of TMDCs, we have performed DFT calculations to reveal the self-intercalation process in terms of both thermodynamics and kinetics. Figure 5a shows the formation energies of self-intercalated NiTe$_2$, PdTe$_2$, and PtTe$_2$ with different concentrations of intercalated Ni, Pd, and Pt, respectively, where negative formation energies are observed for the self-intercalation process of both NiTe$_2$ and PdTe$_2$. Moreover, for the self-intercalation of PdTe$_2$, the formation energy decreases with increasing concentration of intercalated Pd, indicating that the whole self-intercalation process of Pd into PdTe$_2$ is thermodynamically favorable. Although the formation energy of self-intercalating NiTe$_2$ slightly increases with the increase of intercalated Ni, NiTe single crystal can still be obtained by increasing the concentration of Ni ions near the sample surface, which increases the chemical energy of Ni ions. In contrast, for PtTe$_2$ with different concentrations of intercalated Pt, the formation energies are all positive, indicating that spontaneous self-intercalation of Pt into PtTe$_2$ is prohibited at the current approach. Such difference in the formation energy explains why experimentally it is more difficult to achieve a complete phase transition from NiTe$_2$ into NiTe, compared to the intercalation of PdTe$_2$ into PdTe. Considering the kinetics of the self-intercalation process of PdTe$_2$ and NiTe$_2$, the entire ion migration process involves migration from the octahedral interstitial site to the tetrahedral interstitial site and then to the other octahedral interstitial site. The energy profile of the entire migration process shows an "M"-shaped curve, as shown in Fig. 5b, where the migration barriers for self-intercalation of Pd into PdTe$_2$ and Ni into NiTe$_2$ are 0.86 eV and 1.15 eV, respectively. We note that these energies are clearly lower than the value of approximately 1.40 eV for Ta intercalation of 2H-TaS$_2$ bilayer into Ta$_9$S$_{16}$ (ref. 34). Our work provides a convenient route to obtain high-quality TMMCs through ionic liquid gating induced cation self-intercalation, which is complementary to

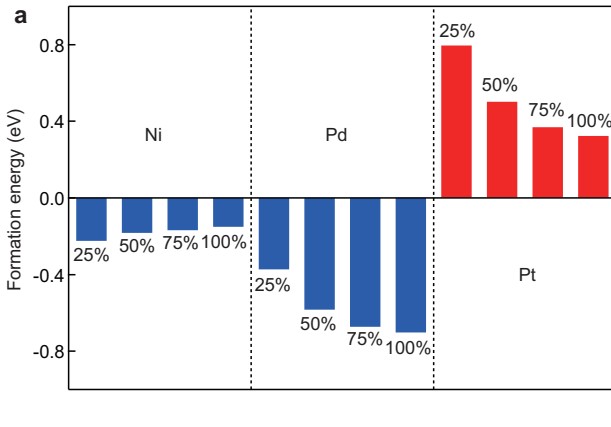

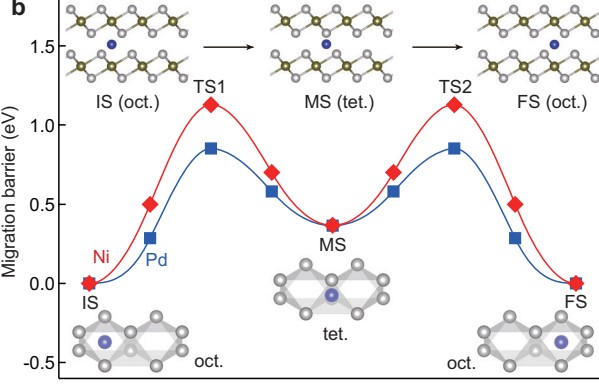

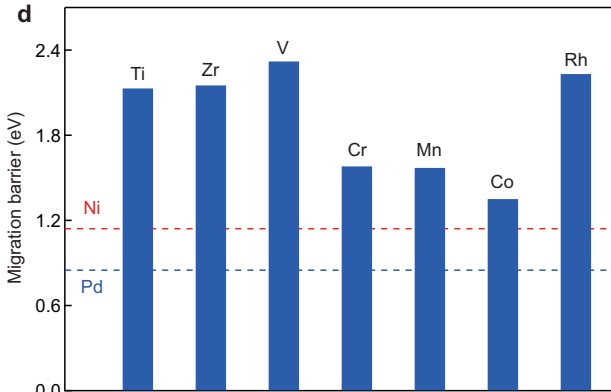

**Fig. 5 | Calculated formation energies and migration barriers of self-intercalation. a** Formation energies of self-intercalated $NiTe_2$, $PdTe_2$, and $PtTe_2$ with intercalation concentrations of 25%, 50%, 75%, and 100%. **b** Energy profiles of the self-intercalation of the Ni and Pd into the $NiTe_2$ and $PdTe_2$. The migration path of ions in the van der Waals gap of TMDCs consists of migration from the octahedral interstitial site to the tetrahedral interstitial site and then to the other octahedral interstitial site. The solid lines represent the migration path, and the symbols represent the intermediate images along the migration path. The IS, MS, FS, and TS represent the initial, intermediate, final, and transition states, respectively. The insets show the top and side view of the different interstitial site configurations. **c** Formation energies of self-intercalated $TiTe_2$, $ZrTe_2$, $HfTe_2$, $VTe_2$, $NbTe_2$, $TaTe_2$, $CrTe_2$, $MnTe_2$, $FeTe_2$, $CoTe_2$, $RhTe_2$, and $IrTe_2$. **d** Migration barriers of the self-intercalation of the Ti, Zr, V, Cr, Mn, Co, and Rh into the $TiTe_2$, $ZrTe_2$, $VTe_2$, $CrTe_2$, $MnTe_2$, $CoTe_2$, and $RhTe_2$, respectively. The red and blue dashed lines represent the migration barriers of the Ni and Pd into the $NiTe_2$ and $PdTe_2$, respectively.

self-intercalation during the molecular beam epitaxy (MBE) growth of thin TMDC films[34].

To investigate the applicability of the self-intercalation method for other TMDCs, we have also performed DFT calculations for additional TMDCs, including Ti, Zr, Hf, V, Nb, Ta, Cr, Mn, Fe, Co, Rh, and Ir. As shown in Fig. 5c, except for HfTe, NbTe, TaTe, FeTe, and IrTe, the formation energies of the remaining seven fully intercalated TMDCs (i.e., TMMCs) are negative, indicating that the self-intercalation of these TMDCs into their corresponding TMMCs is thermodynamically favorable. In addition, Figure 5d shows that the migration barriers for the self-intercalation of Ti, Zr, V, and Rh into their respective $TiTe_2$, $ZrTe_2$, $VTe_2$, and $RhTe_2$ compounds are approximately twice as large as those for $PdTe_2$ and $NiTe_2$. On the other hand, the migration barriers for $CrTe_2$, $MnTe_2$, and $CoTe_2$ are only slightly larger than those for $PdTe_2$ and $NiTe_2$. Therefore, based on both thermodynamic and kinetic considerations, we predict that self-intercalation of $CrTe_2$, $MnTe_2$, and $CoTe_2$ is probably experimentally feasible among the TMDCs we investigated.

In summary, we develop a synthesis method for obtaining PdTe and NiTe single crystals via ionic liquid gating induced self-intercalation of the transition metal ions into TMDCs. Using $PdTe_2$ and $NiTe_2$ as examples, we demonstrate the successful growth of high-quality PdTe and NiTe single crystals, providing opportunities for investigating the properties of these difficult-to-synthesize materials. Isotropic superconductivity with a high critical temperature of 4.3 K

and a higher upper critical magnetic field of 0.25 T at 2 K is reported in PdTe. Further, we elucidate the intercalation mechanism and made further predictions for other possible self-intercalated TMDCs materials. Our results provide a pathway for synthesizing TMMC single crystals, and enriches the applications of ionic liquids in controlling the material phases and properties.

## Methods

### Growth of $PdTe_2$ and $NiTe_2$ single crystals

High-quality $PdTe_2$ and $NiTe_2$ single crystals were synthesized by self-flux method. High purity Pd granules (99.9%, Alfa Aesar) and Te ingot (99.99%, Alfa Aesar) or Ni granules (99.9%, Alfa Aesar) and Te ingot (99.99%, Alfa Aesar) with a molar ratio of 1:9 were loaded in a vacuum-sealed ($10^{-2}$ Pa) silica ampoule. The mixture was heated to 900 °C for 48 h for a better homogenization. Then the mixture was cooled down to 500 °C at 5 °C/h to crystallize $PdTe_2$ or $NiTe_2$ in Te flux. After maintaining at 500 °C for 72 h, the excess Te was centrifuged isothermally, and single crystals of $PdTe_2$ and $NiTe_2$ were obtained.

### Ionic liquid gating

The ionic liquid gating was carried out with a home-designed device. High-quality $PdTe_2$ or $NiTe_2$ single crystal sample was immersed in the ionic liquid electrolyte ($[C_2MIm]^+[TFSI]^-$). The conducting single crystal was clamped by a Pt wire (0.2 mm diameter, bent into a paper clip), which was employed as the working electrode, while a piece of Pt sheet

was grounded and used as the counter electrode. To facilitate the intercalation process, the entire setup was put into a crucible container and placed on a heating plate with the temperature of 150 °C for $PdTe_2$ (170 °C for $NiTe_2$). A negative voltage was then supplied by an electrochemical workstation to the sample, and the voltage was gradually increased to the desired value. The gating process took an extended time of period to achieve fully intercalated samples.

### Scanning transmission electron microscopy (STEM) measurements

The STEM sample was prepared using the focused ion beam (FIB). The sample was thinned down using an accelerating voltage of 30 kV with a decreasing current from 240 to 50 pA, and then with a fine polishing process using an accelerating voltage of 5 kV and a current of 20 pA. Atomic-resolved images and element characterization were acquired using an FEI Titan Cubed Themis G2 300 operated at 300 kV and equipped with a high-brightness Schottky field emission gun and monochromator, a probe aberration corrector to provide a spatial resolution better than 0.6 Å in STEM mode, an energy dispersive EDS detector and a postcolumn imaging energy filter (Gatan Quantum 965 Spectrometer). The HAADF detector's collection angles was 48–200 mrad. The determination of the atomic position was obtained through the statSTEM[35].

### Mechanical exfoliation of NiTe from NiTe/NiTe₂ hetero-junction

NiTe flakes were fabricated from the $NiTe/NiTe_2$ hetero-junctions through repeated mechanical exfoliation using the poly-dimethylsiloxane (PDMS) stamp. We noticed that the bonding strengths at both the interface of $NiTe/NiTe_2$ and at the bulk of $NiTe_2$ are much weaker than that at the interior of NiTe, making it possible to obtain the NiTe flake through exfoliation. To prepare the NiTe flake, a fresh surface of $NiTe/NiTe_2$ sample was first cleaved from the tape and deposited onto the PDMS. After pressing the tape for about 1 min, a flake was then exfoliated onto PDMS. We checked the Raman spectra of this fresh surface, and continued the exfoliation process until the Raman characteristic peak from $NiTe_2$ is not detectable anymore. This together with the well-defined characteristic peak from NiTe strongly suggests that the obtained flake is indeed the single phase NiTe. Afterward, the exfoliated NiTe was aligned and deposited onto the pre-patterned Ti/Au (10 nm/50 nm) Hall bar on Si/SiO₂ substrate under an optical microscope. After heating the substrate at 100 °C for 5 min, the NiTe flake was transferred onto the Hall bar successfully. The characterization of the exfoliated NiTe flake by Raman measurements and the electrical transport measurements are shown in Supplementary Fig. 11.

### Self-intercalation of exfoliated PdTe₂ flake

The exfoliated $PdTe_2$ flakes were deposited on SiO₂/Si substrate with pre-deposited Ti/Au (10 nm/50 nm). Afterward, the substrate was connected with Pt wire by silver epoxy, which was employed as the working electrode, while a piece of Pt sheet was grounded and used as the counter electrode.

### Transport measurements

Transport measurements were performed in a Physical Property Measurement System (PPMS, Quantum Design) with the lowest temperature of 1.6 K and magnetic field up to 9 T. The longitudinal resistance was measured using four-probe method.

### Density functional theory calculations

DFT calculations were performed using the Vienna ab initio simulation package (VASP)[36]. The projector augmented wave (PAW) potential[37] and the generalized gradient approximation (GGA) of the Perdew-Burke-Ernzerhof (PBE) functional[38] were used for the electron-ion interaction and the exchange-correlation energy, respectively. The cutoff energy of the plane wave basis was set to 400 eV. The convergence criteria for the total energy and the force were set to $10^{-5}$ eV and 0.01 eV Å$^{-1}$, respectively. A $19 \times 19 \times 15$ Γ-centered Monkhorst-Pack[39] $k$-mesh was used to sample the Brillion zone. The Heyd-Scuseria-Ernzerhof (HSE06) hybrid functional[40] was used for geometric optimization and total energy calculation to obtain accurate and reliable results. The migration barrier was calculated by using the climbing image nudged elastic band (CI-NEB) method[41,42]. The optB86b-vdW functional[43] was used to describe the van der Waals interaction.

## Data availability

All data supporting the results of this study are available within the article and Supplementary Information. Additional data are available from the corresponding author upon request.

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

## Acknowledgements

This work is supported by the Basic Science Center Program of NSFC (Grant No. 52388201), the Ministry of Science and Technology of China (Grant No. 2021YFA1400100, 2021YFA1400300, 2021YFE0107900), National Natural Science Foundation of China (Grant No. 12234011, 52025024, 92250305). C.B. acknowledges support from the China Postdoctoral Science Foundation (Grant No. 2022M721769). Y.Z. and C.B. acknowledge support from the Shuimu Tsinhua Scholar Program of Tsinghua University.

## Author contributions

S.Z. and P.Y. conceived the research project. F.W. and H.Z. grew, intercalated, and characterized the samples. Y.Z. performed the TEM measurements. F.W., H.Z., and C. B. performed the transport measurements and data analysis. Z.W., X.W., and J.L. performed the first-principles calculations. F.W., Y. Z., Z.W., J.L., P.Y., and S.Z. wrote the manuscript, and all authors commented on the manuscript.

## Competing interests

The authors declare no competing interests.
