## [Peer Review File · Nature Communications]

Ionic liquid gating induced self-intercalation of transition metal chalcogenidesREVIEWER COMMENTS

Reviewer #1 (Remarks to the Author):

Wang et al. reports the experimental method to obtain monochalcogenide single crystals which syntheses have difficulties. They report that ionic liquid gating helps the dissolve and decomposition of the starting dichalcogenide compounds and self-intercalation of the resultant transition metal forms the targeted monochalcogenide. They confirmed the structure and composition of the obtained single crystal by XRD, EDS and STEM. They also confirmed the obtained crystals show the superconductivity and magneto-transport behaviors reproduce the reported behaviors. The DFT calculation indicates the balance of formation energy of monochalcogenide and migration energies, PdTe and NiTe reported here are the easiest to obtain, while possible application is feasible for Co, Mn and Cr tellurides.

The experimental methods are solid, and the results are clear. I believe that the method is very interesting and useful for the fields. The manuscript is worth publishing after minor revision.

There are small questionable comments that are not so clear. Below are some of the examples.

1. In abstract, the abbreviation TMMC appears without the explanation.
2. In page 13 line 8, it is written as “intentionally intercalated with only 100 nm thick NiTe on the surface, and a significant increase of Ni element is observed for the NiTe region.” I wonder what the intention of the authors is here. It seems that they stopped the reaction at some point so that the region of NiTe becomes 100 nm thickness, and it is presented in cross-section STEM, but it is not so clear from the text.

Reviewer #2 (Remarks to the Author):

This manuscript describes how ionic liquid gating can be used to dramatically modify the crystal structure and stoichiometry of some transition metal dichalcogenides. In particular, the authors report on a self-intercalation process, in which transition metal atoms are solubilized and then inserted in the van der Waals gap. As a result, the physical properties of crystals are also profoundly modified.

The authors explore this change in crystal structure in PdTe₂ and NiTe₂. The structural transition and the consequent change in the physical properties of the crystal is clearly demonstrated at least for the case of PdTe₂.

I find this work interesting, as it shows a new way to produce single crystals of transition metal monochalcogenides with high quality.

However, I find some important aspects which should be clarified, especially in the characterization of the NiTe₂/NiTe compounds, as follows:

1. While XRD shows that PdTe₂ crystals undergo a complete phase transition, the self-intercalation of NiTe₂ leaves a large part of the crystal in its pristine state (extended Fig. 6).

This leads to a few questions:

- a. Is it possible to achieve a complete phase transition in NiTe₂ to NiTe? OR a state is reached when the phase transition is not favored anylonger?
- b. To characterize the electrical transport in NiTe, the authors measure on an exfoliated flake. However, NiTe is not a van der Waals material, so I wonder how it is possible to exfoliate it.
- c. Since the bulk crystal is not fully self-intercalated, how are the authors certain that the flake in fig. 4g is NiTe and not NiTe₂? Can the authors show the Raman measured on that very same flake to confirm it is indeed NiTe?
- d. While for PdTe₂ the authors compare the R(T) of pristine and self-intercalated material, for NiTe they only show the MR of the self-intercalated device. The authors should (i) compare it with the MR of NiTe₂, and (ii) show the R(T) of both pristine and intercalated compounds. In this way, differences in the physical properties of the two compounds would appear more clearly (are they both metallic?).
- e. What is the voltage required in the NiTe case to achieve the self-intercalation? In PdTe₂ it is 3.2 V, but for NiTe is not mentioned.

2. Important details of the ionic gating process are not described. As the manuscript is, the

process is hardly reproducible. Is the organic salt dissolved in a solvent? How is the 2D crystal immersed in the solution? Is it attached to another material to be used as an electrode? How is the temperature controlled? I suggest the authors to add a photo of their setup in the SI.

3. From the $M(H)$ of PdTe, the authors can extract the volume (mass) of the crystal which turns superconductive (considering it becomes a perfect diamagnet). Does it correspond to the mass of the measured crystal?

4. The literature is incomplete. Some closely related works on superconductivity and intercalated 2d materials should be cited, such as

<https://doi.org/10.48550/arXiv.2302.05078> and Adv. Funct. Mater. 32, 2208761, 2022.

Overall, this paper shows an interesting phase transition induced by ionic gating, which can be used to obtain high quality single crystals of monochalcogenides. However, the material characterization needs to be improved (especially in the case of NiTe) before I can recommend the publication of this work for Nature communications.

Reviewer #3 (Remarks to the Author):

The manuscript of Wang et al presents interesting results of converting TMDCs to TMMCs using ionic liquid-mediated electrochemical reaction. The results are generally convincing, I would recommend publication after the authors address the following issues.

1) The intercalation process is not fully characterized, are the dissolution and interaction processes happen under the same electrochemical conditions? I would expect different voltages to drive the processes. The IV characteristics during the process should give much information about the electrochemical reactions. Such data should be given and well analyzed.

2) Both processes described in Figure 1c and 1d should happen in this experiment, but the effects and organic molecule interaction are not discussed in the manuscript.

3) The converted PdTe is a 3D bulk crystal, how to understand the anisotropy of H_c under in-

plane and out-of-plane fields, i.e., Figure 3d.

4) I strongly recommend the authors demonstrate that the conversion can happen with 2D PdTe₂, not just bulk crystals. This will be very helpful to further enhance the research novelty, as required by Nature Communication.

5) The MR in Figure 4g is marginal (0.5%), I do not think it's meaningful to conclude that it implies topological properties.

Manuscript NCOMMS-23-16733-T:

We thank all reviewers for valuing the scientific merits of our work and for raising important questions to help us further improve the manuscript. In response to the reviewers' questions, we have performed extensive experiments and revised the manuscript, which have led to major improvement of our manuscript. Below please see a point-by-point response.

Reviewer #1:

Wang *et al.* reports the experimental method to obtain monochalcogenide single crystals which syntheses have difficulties. They report that ionic liquid gating helps the dissolve and decomposition of the starting dichalcogenide compounds and self-intercalation of the resultant transition metal forms the targeted monochalcogenide. They confirmed the structure and composition of the obtained single crystal by XRD, EDS and STEM. They also confirmed the obtained crystals show the superconductivity and magneto-transport behaviors reproduce the reported behaviors. The DFT calculation indicates the balance of formation energy of monochalcogenide and migration energies, PdTe and NiTe reported here are the easiest to obtain, while possible application is feasible for Co, Mn and Cr tellurides. ***The experimental methods are solid, and the results are clear. I believe that the method is very interesting and useful for the fields.*** The manuscript is worth publishing after minor revision.

Reply: We thank the reviewer for the positive comments and suggestions for further improvement of our manuscript.

There are small questionable comments that are not so clear. Below are some of the examples.

1. In abstract, the abbreviation TMMC appears without the explanation.

Reply: We thank the reviewer for pointing out this and we have added the explanation, see page 2 line 31, “transition metal monochalcogenides (TMMCs)”.

2. In page 13 line 8, it is written as “intentionally intercalated with only 100 nm thick NiTe on the surface, and a significant increase of Ni element is observed for the NiTe region.” I wonder what the intention of the authors is here. It seems that they stopped the reaction at some point so that the region of NiTe becomes 100 nm thickness, and it is presented in cross-section STEM, but it is not so clear from the text.

Reply: The referee is correct that the 100 nm thick NiTe (before it is fully intercalated) is employed to obtain further understanding of the intercalation process. With this sample, we were able to carry out cross-section STEM measurements on both pristine and intercalated regions. We find that the phase transformation is driven by the Ni intercalation process, but not due to the Te deficiency, because the EDS analysis shows a significant increase of Ni element from the pristine region (bottom) to new phase region (surface) while the Te signal remains the same. Moreover, the STEM image also confirms that the intercalation starts from surface, with the Ni ions intercalating gradually toward the interior of the NiTe₂ single crystal. It is interesting to note that with the current approach, a vertical NiTe/NiTe₂ hetero-junction is readily achieved, which forms an exotic material platform for future studies.

In response to the reviewer's question, we have revised the discussions to clarify this, please see page 12 line 179, "To understand the intercalation process, we stopped the intercalation process of one sample before it was fully intercalated. The cross-section STEM image (Supplementary Fig. 10) of this sample shows that there is approximately 100 nm thick NiTe on the surface. The EDS analysis shows a significant increase of Ni element while the Te signal remains the same, suggesting that the phase transition is caused by Ni intercalation and not by Te deficiency. Moreover, the STEM image also shows that the intercalation starts from the surface to the interior of the NiTe₂ single crystal, and the obtained NiTe/NiTe₂ hetero-junction forms an exotic material platform for future studies."

Reviewer #2:

This manuscript describes how ionic liquid gating can be used to dramatically modify the crystal structure and stoichiometry of some transition metal dichalcogenides. In particular, the authors report on a self-intercalation process, in which transition metal atoms are solubilized and then inserted in the van der Waals gap. As a result, the physical properties of crystals are also profoundly modified.

The authors explore this change in crystal structure in PdTe₂ and NiTe₂. The structural transition and the consequent change in the physical properties of the crystal is clearly demonstrated at least for the case of PdTe₂.

I find this work interesting, as it shows a new way to produce single crystals of transition metal monochalcogenides with high quality.

Reply: We thank the reviewer for appreciating the scientific merits of our work and suggestions for further improving our manuscript.

However, I find some important aspects which should be clarified, especially in the characterization of the NiTe₂/NiTe compounds, as follows:

1. While XRD shows that PdTe₂ crystals undergo a complete phase transition, the self-intercalation of NiTe₂ leaves a large part of the crystal in its pristine state (extended Fig. 6). This leads to a few questions:

a. Is it possible to achieve a complete phase transition in NiTe₂ to NiTe? Or a state is reached when the phase transition is not favored any longer?

Reply: We thank the reviewer for raising this question. We note that using the current approach, the NiTe₂ sample always transforms into NiTe/NiTe₂ hetero-junction. Since the self-intercalation process is driven mainly the chemical potential difference between the dissolved Te ions within the ionic liquid and that within the bulk layer, we speculate that this could be attribute to the different formation energy between NiTe and PdTe. As revealed by our first-principles calculations (Fig. 5a), the NiTe has a much larger formation energy than that of PdTe, which makes its surface layer act as the rate-limiting factor for the phase transformation. This could nicely explain the fact that a complete phase transition from NiTe₂ into NiTe is more difficult compared to transition from PdTe₂ to PdTe. We note that the intercalation of NiTe₂ leads to NiTe/NiTe₂ hetero-junction, from which single phase NiTe can be obtained through repeated exfoliation. We have added related discussions to the revised manuscript, see page 14 line 205, “Such difference in the formation energy explains why experimentally it is more difficult to achieve a complete phase transition from NiTe₂ into NiTe, compared to the intercalation of PdTe₂ into PdTe.” In addition, experimental details on how to obtain single phase NiTe from NiTe/NiTe₂ hetero-junction through repeated exfoliation is also added to the Methods section “**Mechanical exfoliation of NiTe from NiTe/NiTe₂ hetero-junction. NiTe flakes were fabricated from the NiTe/NiTe₂ hetero-junctions through repeated mechanical exfoliation...**”

b. To characterize the electrical transport in NiTe, the authors measure on an exfoliated flake. However, NiTe is not a van der Waals material, so I wonder how it is possible to exfoliate it.

Reply: As shown in Supplementary Fig. 10, the self-intercalation of NiTe₂ starts from the surface toward the interior of NiTe₂ single crystal, which forms NiTe/NiTe₂ hetero-

junction. We noticed that the bonding strengths at both the interface of NiTe/NiTe₂ and at the bulk of NiTe₂ are much weaker than that at the interior of NiTe, making it possible to achieve the NiTe flake through repeated exfoliation. During the process, we repeated the exfoliation and checked the Raman spectra of the fresh surface every time. We stopped the exfoliation until the Raman characteristic peak from NiTe₂ is not detectable anymore, which together with the well-defined characteristic peak from NiTe strongly suggests that the obtained flake is indeed the single-phase NiTe.

In response to the reviewer's question, we have added details about the exfoliation in methods of the revised manuscript in page 18 line 264, "**Mechanical exfoliation of NiTe from NiTe/NiTe₂ hetero-junction.** NiTe flakes were fabricated from the NiTe/NiTe₂ hetero-junctions through repeated mechanical exfoliation using the polydimethylsiloxane (PDMS) stamp. We noticed that the bonding strengths at both the interface of NiTe/NiTe₂ and at the bulk of NiTe₂ are much weaker than that at the interior of NiTe, making it possible to obtain the NiTe flake through exfoliation. To prepare the NiTe flake, a fresh surface of NiTe/NiTe₂ sample was first cleaved from the tape and deposited onto the PDMS. After pressing the tape for about 1 minute, a flake was then exfoliated onto PDMS. We checked the Raman spectra of this fresh surface, and continued the exfoliation process until the Raman characteristic peak from NiTe₂ is not detectable anymore. This together with the well-defined characteristic peak from NiTe strongly suggests that the obtained flake is indeed the single phase NiTe. Afterward, the exfoliated NiTe was aligned and deposited onto the pre-patterned Ti/Au (10 nm/50 nm) Hall bar on Si/SiO₂ substrate under an optical microscope. After heating the substrate at 100 °C for 5 minutes, the NiTe flake was transferred onto the Hall bar successfully. The characterization of the exfoliated NiTe flake by Raman measurements and the electrical transport measurements are shown in Supplementary Fig. 11."

c. Since the bulk crystal is not fully self-intercalated, how are the authors certain that the flake in Fig. 4g is NiTe and not NiTe₂? Can the authors show the Raman measured on that very same flake to confirm it is indeed NiTe?

Reply: As explained above, single phase NiTe flake can be exfoliated from the NiTe/NiTe₂ hetero-junction. To further confirm that the exfoliated flake is NiTe, we performed Raman mapping measurements on the back side of flake. Since the self-intercalation process occurs from the top surface, the Raman study on the back side of the exfoliated flake can directly confirm whether the exfoliated flake is NiTe. Figure

R1a-c shows the Raman mapping on the back side of an exfoliated flake, in which the characteristic Raman peak at 109 cm^{-1} (Fig. R1c) and the uniform Raman intensity map (Fig. R1b) clearly confirm that the exfoliated flake is indeed NiTe.

Fig. R1. The Raman characterization of NiTe flake. a. The optical photographs of exfoliated NiTe single crystal. b. Raman mapping by integrating the NiTe at 109 cm^{-1} mode. c. Raman spectra from location (1-5) at (b).

In response to the reviewer’s question, we have added the Raman measurements of the exfoliated NiTe flake as Supplementary Fig. 11, and related discussions have also been added to the supplementary information, “Single phase NiTe flake can be exfoliated from the NiTe/NiTe₂ hetero-junction. To further confirm that the exfoliated flake is NiTe, we performed Raman mapping measurements on the back side of flake. Since the self-intercalation process occurs from the top surface, the Raman study on the back side of the exfoliated flake can directly confirm whether the exfoliated flake is NiTe. Supplementary Fig. 11a-c shows the Raman mapping on the back side of an exfoliated flake, in which the characteristic Raman peak at 109 cm^{-1} (Fig. 11c) and the uniform Raman intensity map (Fig. 11b) clearly confirm that the exfoliated flake is indeed NiTe. Supplementary Fig. 11d shows the exfoliated NiTe device for transport measurements, and the resistivity and magnetoresistance (MR) measurements are shown in Fig. 11e,f.”

d. While for PdTe₂ the authors compare the R(T) of pristine and self-intercalated material, for NiTe they only show the MR of the self-intercalated device. The authors should (i) compare it with the MR of NiTe₂, and (ii) show the R(T) of both pristine and intercalated compounds. In this way, differences in the physical properties of the two compounds would appear more clearly (are they both metallic?).

Reply: We thank the reviewer for the suggestions. Figure R2 shows a comparison of the R(T) curves for both intercalated NiTe (Fig. R2a) and pristine NiTe₂ (Fig. R2b) samples, which show that both samples are metallic. Since the current study focuses mainly on the strategy to achieve TMMCs through the self-intercalation process,

detailed studies of the NiTe sample would be carried out for our future work, which would require the experimental conditions of extremely low temperature to testify the proposed superconducting state in NiTe.

Fig. R2. Transport properties of NiTe and NiTe₂ samples. *a, b. Temperature dependent resistance measurements on exfoliated (a) NiTe and (b) NiTe₂ samples.*

e. What is the voltage required in the NiTe case to achieve the self-intercalation? In PdTe₂ it is -3.2 V, but for NiTe is not mentioned.

Fig. R3. Raman spectra at different voltages.

Reply: We thank the reviewer for the careful reading. The voltage used in the NiTe case to achieve the self-intercalation is -3.4 V, and the temperature is 170 °C. This has been added to page 12, line 176, “...and the optimized voltage to achieve the self-intercalation in NiTe₂ is -3.4 V at 170 °C (Supplementary Fig. 9).”

2. Important details of the ionic gating process are not described. As the manuscript is, the process is hardly reproducible. Is the organic salt dissolved in a solvent? How is the 2D crystal immersed in the solution? Is it attached to another material to be used as an electrode? How is the temperature controlled? I suggest the authors to add a photo of their setup in the SI.

Reply: We thank the reviewer for the suggestion. In response to the reviewer’s question, we added the experimental setup as Supplementary Fig. 1 and revised Methods, see page 17 line 245, “**Ionic liquid gating.** The ionic liquid gating was carried out with a home-designed device. High-quality PdTe₂ or NiTe₂ single crystal sample was immersed in the ionic liquid electrolyte ([C₂MIm]⁺[TFSI]⁻). The conducting single crystal connected with Pt wire was employed as the working electrode, while a piece of Pt sheet was grounded and used as the counter electrode. To facilitate the intercalation process, the entire setup was put into a crucible container and placed on a heating plate with the temperature of 150 °C for PdTe₂ (170 °C for NiTe₂). A negative voltage was then supplied by an electrochemical workstation to the sample, and the voltage was gradually increased to the desired value. The gating process took an extended time of period to achieve fully intercalated samples.”

Fig. R4. Experimental setup for the self-intercalation. a. The schematic drawing of the experimental setup. b. Optical image of experimental setup employed for this study.

3. From the M(H) of PdTe, the authors can extract the volume (mass) of the crystal which turns superconductive (considering it becomes a perfect diamagnet). Does it correspond to the mass of the measured crystal?

Reply: As suggested, we used the ZFC M(H) data (Fig. 3e) to calculate the volume of the crystal which turns superconducting. From the mass of PdTe single crystal 1.4 mg, and the density $\rho = 4.6 \text{ g/cm}^3$, the total volume was calculated to be $V = 0.31 \times 10^{-3} \text{ cm}^3$. Then, from the measured total magnetic (m) of $-0.65 \times 10^{-3} \text{ emu}$ at 2 K, the magnetization is calculated to be $M = \frac{m}{V} = -2.1 \text{ emu/cm}^3$. Using the magnetic field of $H = 50 \text{ Oe}$, the diamagnetic susceptibility is calculated to be $\chi = \frac{M}{H} = -0.042$, and the corresponding superconducting shielding volume fraction is equal to $-4\pi\chi = 53\%$.

In response to the reviewer's question, we have added discussion in the revised manuscript in page 11 line 153, "From the diamagnetic response, the superconducting shielding volume fraction is calculated to be 53%."

4. The literature is incomplete. Some closely related works on superconductivity and intercalated 2d materials should be cited, such as <https://doi.org/10.48550/arXiv.2302.05078> and Adv. Funct. Mater. 32, 2208761, 2022.

Reply: We thank the reviewer for pointing out this. These mentioned references have been added as Ref. 27 and 28 in the revised manuscript.

27. Pereira, J. M. et al. Percolating Superconductivity in Air-Stable Organic-Ion Intercalated MoS₂. Adv. Funct. Mater. 2208761 (2022).

28. Wan, Z. et al. Signatures of Chiral Superconductivity in Chiral Molecule Intercalated Tantalum Disulfide. arXiv preprint arXiv:2302.05078 (2023).

Overall, this paper shows an interesting phase transition induced by ionic gating, which can be used to obtain high quality single crystals of monochalcogenides. However, the material characterization needs to be improved (especially in the case of NiTe) before I can recommend the publication of this work for Nature communications.

Reply: We thank the reviewer for the recommendation of publication as well as the constructive suggestions, which have helped us further improve the manuscript.

Reviewer #3:

The manuscript of Wang *et al.* presents *interesting results* of converting TMDCs to TMMCs using ionic liquid-mediated electrochemical reaction. *The results are generally convincing*, I would recommend publication after the authors address the following issues.

Reply: We thank the reviewer for valuing the scientific merits of our manuscript and recommendation for publication.

1) The intercalation process is not fully characterized, are the dissolution and interaction processes happen under the same electrochemical conditions? I would expect different voltages to drive the processes. The IV characteristics during the process should give

much information about the electrochemical reactions. Such data should be given and well analyzed.

Reply: We thank the reviewer for the suggestion. In response to the reviewer's suggestion, we have performed current–voltage (I–V) measurements during the intercalation process and added the cyclic voltametric results as Supplementary Fig. 3, and discussions in line 87 of page7, “This is confirmed by cyclic voltametric measurements of the intercalation process (Supplementary Fig. 2). In order to optimize the experimental conditions, ex-situ X-ray diffraction (XRD), Raman spectroscopy measurements were performed during the intercalation process to monitor the extent of the intercalation at different time durations, from which the optimized experimental conditions for self-intercalation of PdTe₂ are determined to be -3.2 V at 150 °C.” More details have also been added to the supplementary information, “Cyclic voltametric measurements of the intercalation process are shown in Supplementary Fig. 2. The current (or conductivity) starts to increase at the voltage of -1.9 V for PdTe₂ (-1.5 V for NiTe₂) suggesting that the single crystal starts dissolving into the ionic liquid. Upon further increasing the biased voltage to -3.2 V for PdTe₂ (-3.2 V for NiTe₂), the current increases rapidly, indicating that the intercalation reaction is in progress. We propose that the flow of ions leads to a current increase, while some ions will form compounds during this process, which will lead to a decrease in the concentration of conductive ions. When the voltage is beyond -3.6 V for PdTe₂ (-3.6 V for NiTe₂), most of the ions participate in the formation of compounds, so there will be a stage of current decrease. If the voltage continues to increase to -3.8 V for PdTe₂ (-3.7 V for NiTe₂), the sample begins to decompose rapidly.”

Fig. R5. The characterization of intercalation process. Current–voltage (I–V) curves with ionic liquid only (blue curves) and ionic liquid with PdTe₂ sample (a) and NiTe₂ simple (b) at two different temperatures (red curves).

2) Both processes described in Figure 1c and 1d should happen in this experiment, but the effects and organic molecule interaction are not discussed in the manuscript.

Reply: We note that the intercalation of large-sized organic cation ($[C_2Mim]^+$) and Pd/Ni ions would occur at different situations, largely depending on the coupling strength between the layers. For instance, our previous studies have revealed that the intercalation of large-sized organic cation occur at $NdSe_2$, WTe_2 , etc, while the current study clearly suggests that the self-intercalation is favored in $PdTe_2$ and $NiTe_2$. In the later materials, we haven't detected any signature of XRD peaks with larger interlayer spacing, suggesting that the intercalation of large-size organic cations from the ionic liquid was indeed not favored. We deduce that this difference is attributed to the largely different interlayer binding energies between these systems. Previous studies have already demonstrated that $PdTe_2$ has a much larger interlayer binding energy (Figure R6 adapted from Nat. Nanotech. 13, 246–252 (2018)), as compared with other transition metals (such as $MoTe_2$, $NbSe_2$, $SnSe_2$) where organic cation intercalations have already been demonstrated.

In response to the reviewer's suggestion, we have added the discussions in page 8, line 109, "We note that the intercalation of large-sized organic cations such as $[C_2Mim]^+$, which has been reported in $MoTe_2$, $NbSe_2$, $SnSe_2$ etc^{3, 24-27} with characteristic XRD peaks at a larger interlayer spacing, is not observed in the current study, suggesting that the intercalation of large-sized organic cations was not favored in the current study. We believe that this difference could be attributed to the larger interlayer binding energy ($51 \text{ meV}\cdot\text{\AA}^{-2}$) of $PdTe_2$ as compared with that ($\sim 25 \text{ meV}\cdot\text{\AA}^{-2}$) for other TMDCs²⁸. Such larger interlayer binding energy does not favor the intercalation of the large-sized organic cations with dramatic lattice expansion, while the small-sized transitional metal cations can still be intercalated."

Fig. R6. Interlayer binding energy for a selection of layered materials identified in Ref. (Nat. Nanotech. 13, 246–252 (2018)).

3) The converted PdTe is a 3D bulk crystal, how to understand the anisotropy of H_c under in-plane and out-of-plane fields, i.e., Figure 3d.

Reply: We would like to point out that although the converted PdTe is a 3D bulk crystal, this material is still somewhat anisotropic. Figure R7 shows the crystal structures of PdTe from different axis, where the atomic arrangements in the a-b plane and a-c plane are clearly different. Such different structure (bonding environment) of PdTe in a-b plane and a-c plane could explain the observed anisotropy of H_c .

Fig. R7. Crystal structures of PdTe viewed from different directions.

4) I strongly recommend the authors demonstrate that the conversion can happen with 2D PdTe₂, not just bulk crystals. This will be very helpful to further enhance the research novelty, as required by Nature Communication.

Reply: We thank the reviewer for the suggestion. As shown in Fig. R8a, b, the PdTe₂ flake was exfoliated onto Si/SiO₂ substrate, and the thickness is measured to be 170 nm (Fig. R8c, d). Figure R8e-h shows that after 8 hours (at 150 °C, -3.2 V) of ionic liquid

gating, the PdTe₂ flake was successfully transformed into PdTe through intercalation, which is indicated by the characteristic PdTe Raman peak at 98 cm⁻¹ (pointed by red arrow in Fig. R8e and the uniform intensity map shown in Fig. R8g).

Fig. R8. The self-intercalation of PdTe₂ flake. *a.* Optical image of experimental setup for self-intercalation in PdTe₂ flake. *b, c.* The optical image (*b*) and AFM topography (*c*) of exfoliated PdTe₂ flake. *d.* The height of PdTe₂ flake extracted from AFM topography as indicated by the white line in (*c*). *e.* Raman spectra of PdTe₂ flake after different intercalation times. *f.* The optical image of flake after self-intercalation. *g.* Raman intensity mapping by integrating the characteristic mode (at 98 cm⁻¹) of PdTe. *h.* Raman spectra acquired from locations (1-3) at (*g*).

In response to the reviewer’s suggestion, we added the data for self-intercalation of PdTe₂ flake as Supplementary Fig. 4, and discussions in page 7, line 94, “Moreover, the self-intercalation has also been successfully demonstrated on an exfoliated PdTe₂ flake with thickness of 170 nm (Supplementary Fig. 4), showing that the self-intercalation applies not only to bulk crystals but also to thin flakes.” More details have been added to the Supplementary information. “The self-intercalation of exfoliated flake can also be obtained using a similar setup (Fig. 2a). Figure 2b shows an optical image of the exfoliated PdTe₂ flake on Si/SiO₂ substrate, whose thickness is measured to be 170 nm by atomic force microscopy (AFM) measurements (Supplementary Fig. 4c, d). Supplementary Fig. 4e-h shows that after 8 hours (at 150 °C, -3.2 V) of ionic liquid gating, the PdTe₂ flake was successfully transformed into PdTe through intercalation, which is indicated by the characteristic PdTe Raman peak at 98 cm⁻¹ (pointed by red arrow in Fig. 4e) and the uniform intensity map shown in Fig. 4g.”

5) The MR in Figure 4g is marginal (0.5%), I do not think it's meaningful to conclude that it implies topological properties.

Reply: We thank the reviewer for pointing this out. We agree that this MR is not sufficient to conclude the topological properties and we have removed it in the revised manuscript, leaving the investigation of NiTe properties to future studies, see page 14 line 190, “The availability of NiTe single crystals makes it possible to investigate the predicted topological and superconducting properties at low temperature¹⁴ in the future study.”

REVIEWERS' COMMENTS

Reviewer #2 (Remarks to the Author):

The authors have answered all my concerns in a satisfactory way, and I now recommend the publication of this work. I only have a minor doubt about the experimental setup: in the new Supplementary Fig. 1, it is not explicitly said how the 2D crystals are connected to the Pt wire. This is an important detail to reproduce the experiment, so I encourage the authors to clarify it.

Reviewer #3 (Remarks to the Author):

The authors have addressed all my comments with additional data and I recommend its publication in Nature Communication.

One remaining issue is that the experimental details with the flakes intercalation (Fig R8) are missing.

Manuscript NCOMMS-23-16733A:

We thank both reviewers for raising important questions to help us further improve the manuscript. In response to the reviewers' questions, we have added experimental details and revised the manuscript. Below please see a point-by-point response.

Reviewer #2:

The authors have answered all my concerns in a satisfactory way, and I now *recommend the publication of this work*. I only have a minor doubt about the experimental setup: in the new Supplementary Fig. 1, it is not explicitly said how the 2D crystals are connected to the Pt wire. This is an important detail to reproduce the experiment, so I encourage the authors to clarify it.

Reply: We thank the reviewer for the positive comments, in our experiments, we use a Pt wire with a diameter of 0.2 mm. The Pt wire is malleable and easy to bend. We make it into a paper clip to clamp the sample.

In response to the reviewer's question, we have added details in methods of the revised manuscript in page 17 line 241, "The conducting single crystal was clamped by a Pt wire (0.2 mm diameter, bent into a paper clip), which was employed as the working electrode, ...".

Reviewer #3:

The authors have addressed all my comments with additional data and I *recommend its publication in Nature Communications*. One remaining issue is that the experimental details with the flakes intercalation (Fig R8) are missing.

Reply: We thank the reviewer for the recommendation of publication as well as the constructive suggestions. In response to the reviewer's question, we have added the experimental details with the PdTe₂ flakes intercalation in methods of the revised manuscript in page 19 line 273, "The exfoliated PdTe₂ flakes were deposited on SiO₂/Si substrate with pre-deposited Ti/Au (10 nm/50 nm). Afterward, the substrate was connected with Pt wire by silver epoxy, which was employed as the working electrode, while a piece of Pt sheet was grounded and used as the counter electrode."